# Foreign Trade Structure, Opening Degree and Economic Growth in Western China

**Nan Li \*, Lipeng Sun, Xiao Luo, Rong Kang \* and Mingde Jia**

School of Economics and Management, Northwest University, Xi'an 710069, China;
201720144@stumail.nwu.edu.cn (L.S.); lx201720104@stumail.nwu.edu.cn (X.L.); jmd@nwu.edu.cn (M.J.)
\* Correspondence: linan91@stumail.nwu.edu.cn (N.L.); kangrong@nwu.edu.cn (R.K.);
Tel.: +86-188-2920-8551 (N.L.); +86-136-0926-5158 (R.K.)

**Abstract:** This paper presents an interactive study on the relationship between the foreign trade structure, opening degree and economic growth of the provinces in western China (except Tibet). It shows that the export of primary products and labor-intensive products has a positive impact on the external development of the western region, while the export of capital and technology-intensive products has a smaller inhibitory effect on it. At the same time, the system GMM model shows that the opening degree of the western region has a positive effect on economic growth. After including the foreign trade structure interaction item, this result has not changed, and on the basis of opening up to the outside world, the export of labor-intensive products and capital-intensive products plays a significant role in promoting economic development. Therefore, this paper holds that the western region should optimize its foreign trade structure, continue to promote the construction of foreign trade demonstration, and give priority to the development of local characteristic industries to promote economic growth.

**Keywords:** foreign trade; opening degree; economic growth

## 1. Introduction

Since the 1980s, opening up has been recognized as an important economic development strategy of China. China's economic strength and overall national strength have continuously improved, and remarkable achievements have been made. However, as an economic power with a vast territory and a large population, the economic development level difference between regions is increasing day by day, especially in the imbalance of the economic development level between the east and the west, which has become an important issue in the process of China's economic development. It was in the 1980s that, in order to further improve the economic level, China divided the country into three economic zones according to the geographical location of each province: the eastern, central, and western regions. The development model's main aim was to open up the development of the eastern coastal areas to the outside world, and to extend to the inland areas with a gradient-promoting model. The economic development in the eastern region has rapidly increased and the economic strength has significantly improved. However, due to the differences in geographical location and economic strength, there is a great imbalance between the degree of opening up and economic development in the east, the middle, and the west. In order to strengthen the economic development of the western region and gradually narrow the economic gap between the western and eastern regions, China began to implement the strategy of developing the western region in 1999. After more than 20 years of development, the western region has undergone earth-shaking changes and the economic level has achieved rapid development.

Since 2007, the economic growth rate of the western region has surpassed the eastern region in 11 consecutive years, and is in the period of medium-to-high speed development opportunities. In terms of the total amount, the GDP of the western region increased from 12.7 trillion to 17.1 trillion yuan from 2013 to 2017, accounting for 20% of the country's total, with an average annual growth rate of 8.8%. In terms of provincial development, the GDP growth rates of Guizhou, Tibet, and Yunnan were the top three rates across the countries in 2017, the first two of which are the only provinces in China to achieve double-digit GDP growth. The development and opening up of the western region has always been the focus of development, but the degree and level of opening up to the outside world is relatively low, which is still the "short board" in the layout of China's regional opening up. In terms of foreign trade and investment, the western region accounted for only 6.8% of the total foreign trade, 10% of the usage of foreign capital, and 7.7% of the total foreign investment in 2017. The export products are mainly transformed from resource-intensive to labor-intensive, and the export of machinery and electrical equipment has grown rapidly, while its international competitiveness is not strong. The development of the western region in the future will focus on further opening up. The promotion of the Strategy of Developing the Western Regions, implementation of the "the Belt and Road" initiative and establishment of the third batch of pilot free trade zones have generated significant opportunities and policy support for the economic development of the western region. Foreign trade is an important part of regional economic development. Regions with faster development of foreign trade are also regions with the highest level of economic development. In order to realize the leap-forward economic development in western China, we must make full use of the supporting role of opening up to the outside world in regional economic development, further optimize the structure of foreign trade, and give full play to the comparative advantages and regional characteristics of the western provinces.

Due to the high level of opening up to the outside world and economic development in the eastern region, domestic and foreign scholars have developed deep research knowledge on it, and significant literature has been published from various perspectives. However, due to the widening gap between the western region and the eastern one, people seem to pay little attention to the internal differences in the western region except for the overall concern of the western region. In fact, the western region is a multi-ethnic region. The economic, cultural, and social developments of the 12 western provinces and cities have their own characteristics. The opening situation and the foreign trade structure are even more different. Therefore, based on the foreign trade structure and the degree of openness in the western region (except Tibet), this paper discusses their impact on the economic development of the western region. It also makes a comprehensive analysis of the changing trend of foreign trade structure and the development of foreign trade transformation demonstration bases in the western regions in order to realize high-quality economic development in the western regions.

## 2. Literature Review

Foreign scholars' research on the relationship between export commodity structure, opening degree, and economic growth includes the following studies. McNab and Moore (1998) conducted an empirical study on the data of 41 countries and found that export-oriented trade policies had a significant positive impact on the economic growth of developing countries by directly promoting exports. Kenani and Fujio (2012) analyzed the relationship between financial development, trade openness and economic growth in Malawi based on the VAR (Vector Autoregression) model and found that there was a long-term positive correlation between them. Sheridan (2014) used a sample of the endogenous technology division to determine the level of manufacturing exports and GDP per capita growth possible economic development between the threshold. The research results showed that a country needs to reach the lowest level of human capital to benefit from dependence on commodity exports to make the transition to manufacturing exports, which means that a threshold value of horizontal exports of manufactured goods can result in higher returns. Sakyi et al. (2015) found that there is a positive bidirectional relationship between trade openness and income level in the long-term. At the same time, they investigated that the trade openness has had an impact on the levels of income

and rates of growth in a sample of 115 developing countries for 1970–2009. Pradhan et al. (2017) used a panel vector error-correction model (VECM) to study the identification of trade openness, foreign direct investment, financial development, and economic growth in 19 Eurozone countries over 1988–2013. And the results showed that increased inflows of foreign direct investment in the short-term have propelled economic growth, which has strengthened financial development and international trade to sustain economic growth. Ahmad et al. (2018) used a three-stage procedure based on unit root, co-integration and causality tests to analyze the relationships between exports, FDI and economic growth among the ASEAN5 countries. And the results revealed that there is a bi-directional causal relationship between FDI and growth in the long-term, while there is a unidirectional trend from FDI to exports in the short-term. Moreover, Huchet et al. (2018) studied the causality between openness, growth and income distribution by proposing a more elaborated way of measuring openness. By taking into account two additional dimensions of countries' integration in world trade: quality and diversification, they found that the trends of exporting higher quality products and more diversified products grow more rapidly.

Domestic scholars mainly study the relationship between export commodity structure, opening degree, and economic growth from two aspects. First, they study the impact of export commodity structure and trade mode on economic growth. For instance, Dai and Hu (2018), studied the relationship between China's export commodity structure and economic growth. They found that the export of primary and industrial products had a positive impact on economic growth. At the same time, they distinguished the different effects of different commodity types on the economy in the long and short-term. On the relationship between foreign trade development, FDI and economic growth, some scholars draw the conclusions that commodity structure can promote economic growth significantly. At the same time, for different economic growth modes, the impact of import and export trade on the transformation of economic growth mode is different (Zhao et al. 2015; Hu 2018). Zhao (2014) documented the relationship between foreign trade structure, industrial structure and economic growth. He believed that industrial structure and foreign trade structure are the main manifestations of a country's internal and external economy, which have a direct impact on economic growth. Therefore, by sorting out the relevant literature, this paper summarizes the current situation of research, and lays a foundation for future research. Xu (2010) used the export extension model to study the impact of different trade modes on China's economic growth, and concluded that the growth of general trade exports can significantly promote China's economic growth, while the growth of processing trade imports and exports can significantly weaken China's economic growth. Shao and Liu (2011) used the classical OLS (Ordinary Least Square) method and spatial econometrics analytical method to study the relationship between the export structure and the growth rate of per capita GDP of 222 cities above the prefecture level in China from 2001 to 2008. It was found that there was a significant correlation between export concentration and regional economic growth. The higher the degree of specialization of export structure, the higher the growth rate of per capita GDP. Liu (2011) measured the growth effect among Jiangsu's product structure, mode of trade, and export trade, and found that the export of capital-technology-intensive products in Jiangsu Province had a significant positive effect on economic growth, while the export of labor-intensive products had a negative effect at the beginning, and the export of industrial manufactured products could generate a greater economic effect than that of primary products. Lv (2012) used the spatial panel data model to test the economic growth rate and structural factors of the eastern, central, and western regions since China's Reform and Opening, and the results show that the economic growth of the eastern region has been in the stage of "structural deceleration", the western region is in the stage of "acceleration", and the central region coexists with "acceleration" and "deceleration". Song and Chen (2013) argued that exports only have a pulling impact on the economic growth of western China in the short term, while imports have a strong and sustained impact on GDP. Zhang and Zhong (2015) found that export can promote economic growth on the whole in a study on the relationship between export trade patterns and regional economic growth at the provincial level during 1998–2013. However, due to regional differences, exports have

the greatest promotion effect on the eastern region, then the central region, and finally the western region. Dong and Zhao (2017) put forward that the optimization and adjustment of export trade structure are important ways to achieve high-quality economic development of a country.

Secondly, scholars study the impact of the degree of openness on economic growth. Some have taken all provinces of China as the research objects and believed that the degree of openness can promote economic growth (Mao and Sheng 2012); Xu and Liu 2013; Dong and Yan 2015). Some have taken provincial panel data to empirically analyze the relationship between opening up, economic growth and urban-rural income gap, and found that the improvement of China's overall opening-up level can significantly promote regional economic growth. From the regional sample, the comprehensive opening up of the eastern regions or the developed regions plays a stronger role in promoting economic growth, but the effect on narrowing the urban-rural income gap is not obvious. The effect of comprehensive opening up in the central or western regions or underdeveloped regions on narrowing the income gap between urban and rural areas is more obvious, but the effect on regional economic growth is weaker (Zhang 2017; Liu 2018; Li 2019). Some used dynamic panel regression to empirically examine the effects of market-oriented reform and opening up in China's regional economic growth, and found that the marketization process has a significant promoting effect on economic growth. However, with the basic establishment of the market economic system, the growth promotion effect of the marketization process has a decreasing trend. Economic openness is closely related to the level of economic development, especially the improvement of economic openness in eastern China, has effectively promoted the local economic development (Liang 2017; Wang 2018; Jiang 2019). From the perspective of China's regional economy, some have concluded that there are obvious differences among regions through the study of the relationship between the degree of openness and economic growth in China's three major regions, the reason for which is different degrees of opening up (Lan 2002; San 2002). Luo (2007) found that the degree of influence of the opening degree on economic growth in different regions of the eastern coastal areas is different, showing that the Pearl River Delta has the lowest elasticity to economic growth, the Bohai Rim has the highest elasticity, and the Yangtze River Delta is between the two by an empirical analysis of the opening degree and economic growth of the eastern coastal areas. Additionally, some have drawn the conclusion that the opening up of the western region contributes to economic growth based on the Strategy of Developing the Western Regions (Zhao 2017).

To sum up, academia has completed much in-depth research on the above two topics, but little research has been conducted on the relationship between foreign trade suture, opening degree, and economic growth at the same time, especially the relationship between foreign trade structure and openness. In recent years, with the implementation of the Strategy of Developing the Western Regions, the economy in the western region has achieved rapid growth. The level of opening up and high-quality development of the economy in the western region have always been focused on by all social sectors. In view of this, this paper aims to explore how the western region can achieve high-quality economic development on the basis of opening-up and the adjustment of foreign trade structure through the empirical analysis of the relationship between foreign trade structure, opening degree and economic growth in the western region.

## 3. Methodology

### 3.1. Model Specification

The change of foreign trade structure has different impacts on opening up. Discussing the change of the foreign trade structure in western China can better demonstrate the development trend of foreign trade and provide clear ideas for us to consider when discussing regional opening. In this study, we modeled the impact of export changes of primary products, labor-intensive products, and capital-technology-intensive products on the opening degree, and determined the relationship between variables:

$$fdir_{it} = \alpha_0 + \alpha_1 X_{it} + \sum Control_{it} + \mu_i + \sigma_t + \varepsilon_{it} \tag{1}$$



where $\alpha_0$ is the constant coefficient; $X_{it}$ is the core explanatory variable, representing the export proportion of primary products, labor-intensive products, and capital-technology-intensive products; $\alpha_1$ is the explanatory variable coefficient; $Control_{it}$ is the control variable affecting the opening degree; $\mu_i$, $\sigma_t$ are the fixed effects of provinces and years, and $\varepsilon_{it}$ is a random perturbation term.

There are many factors that can be used to promote the economic growth of a region. This paper studies the relationship between the two from the perspective of the opening degree. Since the current economic growth will be affected by the previous level of economic growth, this paper introduces the historical level of economic development for the dynamic analysis of economic growth. Here, the system GMM is used to construct the dynamic model of the impact of the opening degree on economic growth, so as to measure the relationship between the two in a more scientific way.

$$
\begin{aligned}
lnGDP_{it} = \beta_0 \quad &+\beta_1 lnGDP_{i,t-1} + \beta_2 lnGDP_{i,t-2} + \beta_3 lnGDP_{i,t-3} \\
&+\beta_4 lnGDPi_{,t-4} + \beta_5 fdir_{it} + \sum Control_{it} + \mu_i + \varepsilon_{it}
\end{aligned}
\tag{2}
$$

Here, $lnGDP_{i,t-1}$, $lnGDP_{i,t-2}$, $lnGDP_{i,t-3}$, and $lnGDP_{i,t-4}$ represent the lagging one-stage, lagging two-stage, lagging three-stage, and lagging four-stage variables of economic growth, respectively; $fdir_{it}$ is the core explanatory variable and $Control_{it}$ is a series of control variables. In this paper, it is believed that tourism income, railway operating mileage, the technology import cost, and total exports will have a certain impact on regional economic development, so these variables are added here as controls. After systematic GMM regression, this study also conducted sequence autocorrelation and an over-identification test, and all models passed the test. Therefore, this paper has reason to believe the rationality of modeling.

Based on the study of the relationship between opening up to the outside world and economic growth, this paper further introduces foreign trade structure and tries to discuss the influence of the change of foreign trade structure on economic growth, on the basis of opening up to the outside world. The purpose of this it to better grasp the interaction effect of opening up to the outside world and foreign trade structure on economic growth. Therefore, the interaction term of opening-up and foreign trade structure are introduced to construct the interaction terms and economic growth relationship model:

$$
\begin{aligned}
lnGDP_{it} = \gamma_0 + \quad &\gamma_1 lnGDP_{i,t-1} + \gamma_2 lnGDP_{i,t-2} + \gamma_3 lnGDP_{i,t-3} + \gamma_4 lnGDP_{i,t-4} \\
&+\gamma_5 fdir_{it} * X_{it} + \sum control_{it} + \mu_i + \varepsilon_{it}
\end{aligned}
\tag{3}
$$

Here, $lnGDP_{i,t-1}$, $lnGDP_{i,t-2}$, $lnGDP_{i,t-3}$, and $lnGDP_{i,t-4}$ represent the lagging one-stage, lagging two-stage, lagging three-stage, and lagging four-stage variables of economic growth, respectively; $fdir_{it}*X_{it}$ represents the interaction term between opening up and foreign trade structure; and $Control_{it}$ is a series of control variables. Similarly, the system GMM is used for dynamic relationship regression.

### 3.2. Data and Variables

The data of this paper include data collected from the western provinces between 2006 and 2016. Given the lack of data in Tibet, this paper only covers 11 provinces (cities) in the west, and those data were obtained from the Statistical Yearbook, Statistics Bureau, and Business Department of each province (city). The reason why the research time selected was from 2006 to 2016 is that, since China's accession to WTO (World Trade Organization), the degree of economic globalization has further deepened, and China's dependence on foreign trade has increased rapidly. In 2002, it exceeded 50%, increased to 63% in 2005, and even reached the highest figure of 67% in 2006. Since then, it has been deeply affected by China's economic transformation, structural adjustment of internal and external demand, and financial crisis. From 2007, the dependence on foreign trade has gradually declined. In addition, since 2006, China's transition period agreed upon in the WTO negotiations has basically ended; that is, the highest level of market opening has been achieved, all non-tariff measures have been be eliminated, and most product tariffs have fallen to the promised end point for a long time.

Based on the above two points, the authors believes that 2006 is a dividing line for western regions, so the data of this paper are selected from 2006.

The core variables in this paper are the structure of foreign trade, opening degree, and economic growth. The structure of foreign trade is divided by the proportion of different trade classifications amounts in the total amount. The trade classification of this paper, drawing on the articles of some scholars (Song 2013; Fu and Chen 2006), is divided according to the Standard International Trade Classification (SITC) principle of the United Nations, and we further divide the SITC into three categories, which are primary products, labor-intensive products, and capital and technology-intensive products, respectively. The opening degree is measured by the degree of utilization of foreign capital, and the economic growth is measured by the gross domestic product. The specific variable information is shown in Table 1.

**Table 1.** Variable information.

| Name of Variable | Abbreviation of Variable | Standard of Measurement |
|---|---|---|
| primary product share | primary | primary exports/total exports |
| labour intensive product share | labour | export of labour intensive products/total exports |
| capital technology intensive products share | capital | capital technology intensive exports/total exports |
| opening degree | fdir | foreign investment in China/GDP |
| economic growth | lnGDP | gross domestic product |
| tourism income | tourist | one hundred million yuan |
| number of foreign invested enterprises | fdifirm | ten thousand each |
| investment in fixed assets of the whole society | lnfixedinvest | one hundred million yuan |
| urbanization level | urban | urban population/Total population |
| industrial structure | industry | added value of the secondary industry/GDP |
| railway mileage | railway | ten thousand kilometers |
| technology import cost | techimport | one hundred million yuan |
| total exports | export | one hundred million yuan |

As can be seen from the descriptive statistical results in Table 2 (supplementary data Table S1), the variance value of the variables selected in this paper is relatively small, but the variance value of tourism income is relatively large. This paper holds that the main reason for this is the regional historical, geographical characteristics and the comprehensive environmental differences. Therefore, the variables selected in this paper can be considered appropriate variables.

**Table 2.** Descriptive statistical results of variables.

| Variable | Average Value | Variance | Minimum Value | Maximum Value |
|---|---|---|---|---|
| primary | 13.2604 | 20.0511 | 0.5394 | 91.0393 |
| labour | 34.9184 | 27.8974 | 0.1931 | 94.6660 |
| capital | 51.8211 | 29.3916 | 0.0705 | 93.0915 |
| fdir | 1.2657 | 1.5536 | 0.0386 | 9.99720 |
| lnGDP | 8.7680 | 0.8778 | 6.4746 | 10.3945 |
| tourist | 1253.5320 | 1378.8240 | 0.7400 | 7705.5000 |
| railway | 0.3345 | 0.2221 | 0.0800 | 1.2100 |
| fdifirm | 0.3147 | 0.2672 | 0.0123 | 1.205 |
| lnfixedinvest | 8.5144 | 0.9671 | 6.0126 | 10.2686 |
| urban | 44.4872 | 8.0487 | 27.4526 | 62.5984 |
| Industry | 47.2554 | 5.2816 | 34.9365 | 58.3786 |
| techimport | 4.8809 | 7.2872 | 0.0000 | 37.6013 |
| export | 999.3563 | 1857.093 | 17.1594 | 10836.54 |

## 4. Results and Discussion

### 4.1. Analysis of the Influence of Foreign Trade Structure on Opening Degree

Through the analysis of the model regression results, it can be seen that the second, fourth, and sixth columns in Table 3 are the regression results without adding control variables, and the third, fifth, and seventh columns are the regression results after adding control variables. It can be seen that the export of primary products has a positive but not significant impact on the openness of the western region, and the coefficient value increases after adding control variables. Labor-intensive products in the western region have a significant positive impact on the opening degree. We can see that the tourism industry also has a significant positive impact. Furthermore, companies with foreign investment and industrial structure promoted the opening up to the outside world, but not significantly, and the reason for this may be that the attraction of foreign investment in the western region is low, so the rate of investment and construction of factories in the western region is low and the impact on the opening up is weak. Capital and technology intensive products have a negative impact on opening up. The possible reasons for this are the slow development of the financial industry in the western region, the relatively lagging level of technological innovation caused by the low degree of capital concentration, and the fact that the export of technology-intensive products has not been recognized. On the contrary, the western region has been relying on the development of labor-intensive products, due to the accumulation of historical experience in development, and it can achieve specialization, thus improving the openness.

**Table 3.** Impact of foreign trade structure on opening degree.

| Variable | fdir | fdir | fdir | fdir | fdir | fdir |
|---|---|---|---|---|---|---|
| primary | 0.001 (0.15) | - | - | 0.009 (1.25) | - | - |
| labour | - | 0.053 *** (4.11) | - | - | 0.041 *** (4.08) | - |
| capital | - | - | −0.033 *** (−3.47) | - | - | −0.026 * (−3.47) |
| tourist | - | - | - | 0.001 *** (3.21) | 0.001 ** (2.47) | 0.001 *** (2.97) |
| fdifirm | - | - | - | 1.829 (0.97) | 1.809 (1.13) | 1.926 (1.13) |
| lnfixedinvest | - | - | - | 0.125 (0.23) | −0.675 (−1.10) | −0.254 (−0.45) |
| urban | - | - | - | −0.160 (−1.53) | −0.128 (−1.17) | −0.146 (−1.39) |
| Industry | - | - | - | 0.048 (1.20) | 0.052 (1.42) | 0.053 (1.40) |
| Province fixed effect | YES | YES | YES | YES | YES | YES |
| Year fixed effect | YES | YES | YES | YES | YES | YES |
| _Cons | −23.645 (−0.69) | 63.45 * (1.73) | 9.828 (0.26) | 29.925 (0.04) | −324.93 (−0.87) | −154.13 (−0.47) |
| N | 121 | 121 | 121 | 121 | 121 | 121 |
| $R^2$ | 0.52 | 0.65 | 0.60 | 0.65 | 0.71 | 0.69 |

Notes: *, ** and *** mean significant at 10%, 5% and 1% respectively; The t-statistic is in parentheses.

### 4.2. Evolution of Dynamic Relationship between Opening-Up and Economic Growth

As can be seen from Table 4, the lag period of economic growth has a significant impact on the current economic growth. Lag phase i and lag phase iii have positive effects, while lag phase ii and lag phase iv have negative effects. At the same time, the opening degree has a positive impact on economic growth. Whether control variables are added or not, these positive effects are significant. It can be seen that the opening up of the western region has greatly promoted the local economic development. Thanks to the implementation of the central government's policy of further opening up to the outside world, the requirements for a deeper and higher level of opening-up, and coupled with the implementation of the "the Belt and Road" initiative, the western region's links with the outside world have continued to strengthen. These powerful measures have stimulated the economic vitality of the western region and promoted the development of the local economy. In addition, tourism and technology introduction have a positive impact on economic growth, and tourism development has a significant positive impact on economic growth. The influence of railway mileage development and export on economic growth is negative, but not significant.

**Table 4.** The dynamic relationship between opening-up and economic growth.

| Variable | lnGDP | lnGDP |
|---|---|---|
| lnGDP(L1) | 1.116 *** (10.74) | 0.985 *** (7.58) |
| lnGDP(L2) | −0.336 *** (−4.16) | −0.266 *** (−3.05) |
| lnGDP(L3) | 0.282 *** (6.84) | 0.238 *** (5.74) |
| lnGDP(L4) | −0.206 *** (−4.66) | −0.157 ** (−2.49) |
| _cons | 1.336 *** (4.52) | 1.869 *** (4.88) |
| fdir | 0.018 *** (3.04) | 0.015 * (1.96) |
| tourist | - | 0.00002 *** (2.64) |
| railway | - | −0.141 (−0.70) |
| techimport | - | 0.001 (0.86) |
| export | - | $-9.76 \times 10^{-6}$ (−0.55) |
| N | 77 | 77 |
| AR(2) *p*-value | 0.426 | 0.836 |
| Sargan *p*-value | 0.999 | 1.000 |

Notes: *, ** and *** mean significant at 10%, 5% and 1% respectively; The t-statistic is in parentheses.

### 4.3. An Analysis of the Dynamic Relations between Opening-Up, Trade Structure and Economic Growth

At the beginning, this study analyzed the impact of trade structure on opening up and the role of opening up on economic growth, respectively. Through the analysis of trade structure and opening-up, the final aim of this paper is to explore how the western region of China can promote economic development with a higher quality. This paper holds that the positive role of opening up to the outside world in promoting economic growth is closely related to the change of trade structure, and under the implementation of the policy of opening up to the outside world, the effect of the change of trade structure on economic growth is worth paying attention to. Therefore, this paper attempts to discuss the interaction between opening up and trade structure on economic growth. Therefore, on the basis of the dynamic impact of opening up on economic growth, this paper introduces the interaction item of foreign trade structure in order to clearly grasp the relationship between the three variables.

From Table 5, we can see that the lag period of economic growth has a significant impact on the current economic growth, irrespective of whether the control variables are added, which can explain the rationality of the dynamic model in this paper. The lagging periods i and iii of economic growth have positive effects on the current economic growth, while the lagging periods ii and iv have negative effects on the current economic growth. Furthermore, without adding control variables, the interaction between opening-up and foreign trade structure has a positive impact on economic growth. Among them, the interaction between opening-up and labor-intensive exports, and the interaction between opening-up and capital-technology-intensive exports have a significant impact on economic growth. After adding control variables, this result has not changed, which indicates that the export of labor-intensive products and capital-intensive products is good for the regional economy development. The results also pass the AR test and Sargan test, which proves the scientificity of the results.

### 4.4. A Summary of the Relations between Foreign Trade Structure, Opening-Up and Economic Growth

Based on the above results, the export of primary products and labor-intensive products contributes to the improvement of the degree of opening-up, and the latter has a more significant impact on the opening-up. Due to the relatively backward technological level and low concentration of finance and capital in the western region, it is a reasonable choice for the western region to develop primary products and labor-intensive products. Primary products and labor-intensive products have acted as the support for the development of the western region. The western region, with lower labor costs and industry in the threshold, was lower than those of the east, so the corresponding industries are located in the west. On the one hand, this has solved a large number of employment problems, but on the other hand, this has also accelerated the western external ties and regional economic development.

**Table 5.** Dynamic relations among opening-up, trade structure and economic growth.

| Variable | lnGDP | lnGDP | lnGDP | lnGDP | lnGDP | lnGDP |
|---|---|---|---|---|---|---|
| lnGDP(L1) | 1.126 *** (16.04) | 1.178 *** (12.60) | 1.149 *** (13.78) | 1.052 *** (10.39) | 1.079 *** (9.68) | 1.151 *** (12.83) |
| lnGDP(L2) | −0.317 *** (−4.00) | −0.359 *** (−4.67) | −0.320 *** (−4.8) | −0.294 *** (−3.35) | −0.306 *** (−3.61) | −0.309 *** (−4.53) |
| lnGDP(L3) | 0.339 *** (5.52) | 0.323 *** (5.02) | 0.229 *** (5.30) | 0.308 *** (5.83) | 0.311 *** (6.19) | 0.186 *** (3.91) |
| lnGDP(L4) | −0.279 *** (−7.19) | −0.266 *** (−3.86) | −0.177 *** (−5.38) | −0.227 *** (−4.06) | −0.204 *** (−3.18) | −0.221 *** (−7.54) |
| _cons | 1.226 *** (5.96) | 1.148 *** (3.39) | 1.102 *** (5.05) | 1.539 *** (5.37) | 1.213 ** (2.32) | 1.666 *** (5.02) |
| X1(fdir*primary) | 0.0004 (1.35) | - | - | 0.0003 (1.08) | - | - |
| X2(fdir*labour) | - | 0.0002 *** (3.30) | - | - | 0.0002 * (1.71) | - |
| X3(fdir*capital) | - | - | 0.001 *** (3.18) | - | - | 0.0004 *** (3.61) |
| tourist | - | - | - | 0.00002 ** (2.38) | $5.40 \times 10^{-6}$ (0.40) | 0.00003 *** (4.96) |
| railway | - | - | - | −0.088 (−0.56) | −0.136 (−0.70) | 0.064 (1.56) |
| techimport | - | - | - | 0.001 (1.36) | 0.002 ** (2.41) | 0.001 (0.97) |
| export | - | - | - | −0.00003 (−1.21) | −0.00002 (−0.96) | −0.00002 (−1.18) |
| N | 77 | 77 | 77 | 77 | 77 | 77 |
| AR(2) *p*-value | 0.576 | 0.809 | 0.795 | 0.423 | 0.897 | 0.281 |
| Sargan *p*-value | 0.999 | 0.999 | 0.999 | 1.000 | 1.000 | 1.000 |

Notes: *, ** and *** mean significant at 10%, 5% and 1% respectively; The t-statistic is in parentheses.

The expansion of openness also has a positive impact on economic growth. The expansion of openness to the outside world has also had a positive impact on economic growth. Expanding the opening up of the western region to the outside world and strengthening the contact between the western region and the outside world will help build a broader platform and provide more development opportunities for the western region, thus promoting the economic development of the western region with a higher quality. Most of the provinces in the western region are inland. With the implementation of the "the Belt and Road" initiative and opening of trains for trade between China and Europe, China's relations with central Asia, west Asia, and Europe have been increasingly strengthened and the opening up to the outside world has reached a new height. This has also provided a favorable external environment for the economic development of the western region.

For the western region of China, the foreign trade structure plays a unique role in the opening up to the outside world, and opening up to the outside world has a positive impact on regional economic growth. With regards to opening up to the outside world, how to achieve high-quality development of the regional economy is the focus of this paper. Therefore, on the basis of the influence of opening up to the outside world on regional economic growth, we introduce the factor of foreign trade structure to grasp the effect of changes in foreign trade structure on economic growth. The result shows that the foreign trade structure has strengthened the role of opening up to the outside world in promoting regional economic growth, especially in the export of labor-intensive products and capital-technology-intensive products, which is particularly significant. It shows that while insisting on expanding opening-up, the western region should also continue to take advantage of the export of labor-intensive products with regional characteristics. At the same time, it should further enhance its attraction to finance and capital, enhance its ability of scientific and technological innovation, and create a good gathering environment for the development of capital-intensive products, so as to promote high-quality economic development of the region while deepening the opening-up.

*4.5. Further Discussion*

4.5.1. The Trend of Foreign Trade Structure in Western China

This paper attempts to further explore the development of foreign trade structure in the western region in order to find out the trend of foreign trade structure changes in the western region, so as to provide more comprehensive predictions and suggestions for the opening up and economic development of the western region. In 1999, the central government began to implement the strategy of developing the western region. This is a relatively profound transformation for the western region, and means that the central level began to narrow the gap between the central and western regions and improve the speed and quality of economic development in the western region. Therefore, the analysis of the development of foreign trade structure in the western region includes data from 1999 to 2016.

The development trend of the three major trade categories of 11 provinces (cities) in the western region from 1999 to 2016 can be seen from the relevant data. Inner Mongolia, Shaanxi province, Ningxia Hui Autonomous Region, Chongqing city, and Guangxi Zhuang Autonomous Region have a higher proportion of capital and technology-intensive products, indicating that the attraction of technology and capital in these regions is stronger than in others. The larger proportion of labor-intensive products in Qinghai province and Xinjiang Uygur Autonomous Region reflects the advantages of the product-processing industry. In general, the proportion of resource-intensive industries in the 11 regions is lower, while the labor-intensive industries and capital-intensive industries are more active. It can be concluded that the distribution proportion of the trade development pattern in the western region is more reasonable. However, in the long run, labor costs are constantly increasing and the western region will eventually lose the advantage of cheap labor. Capital-intensive products have a high added value and result in a low level of pollution. Therefore, all provinces and cities should develop capital-intensive products, improve the quality of the products, and gain more market recognition.

### 4.5.2. Summary of Demonstration Bases for Foreign Trade in the Western Region

In order to promote foreign trade and consolidate and upgrade China's status as a major trading country, in 2011 the Ministry of Commerce released the Overall Plan for the Cultivation of the Transformation and Upgrading of the Demonstration Base for Foreign Trade of the Ministry of Commerce and the 2011 National Work Plan for the Identification of Professional Model Bases for Transformation and Upgrading of Foreign Trade, designed to promote the establishment of a national demonstration base for foreign trade. The selection of demonstration bases generally requires that they have certain demonstration-driven effects, including industry-specific agglomeration capabilities, leading production enterprises, and economic development zones or special supervision zones established by the government. Therefore, drawing on the Western Blue Book: China's Western Development Report (Yue et al. 2018) and other relevant contents, a summary of demonstration bases for foreign trade in the western region can assist in developing a relatively clear understanding of advantageous industries in the western region and the future direction of development, and provide references for industries in the future.

As can be seen from Table 6 and Figure 1, the number of national foreign trade demonstration bases in Inner Mongolia, Sichuan, Chongqing, and Guangxi is large, and the ten bases in Inner Mongolia far exceeds the number in other provinces and cities, reflecting the high expectations for the development of foreign trade. Furthermore, as can be seen from Table 5 and Figure 2, the industries on the list of bases are mainly concentrated in the agricultural products, textiles and clothing, new materials, and pharmaceutical industries. In general, the key support for foreign trade relies on the development of agricultural products with special local characteristics, such as liquor in Guizhou, flowers in Yunnan, mustard in Chongqing, apples in Shaanxi and Gansu, beef in Inner Mongolia, and so on, which means that localities take advantage of comparative advantages when engaging in foreign trade. With the implementation of the "B&R" (the Belt and Road) the strategy and access of all western provinces to foreign markets, enterprises should use this advantage and increase the added value of agricultural products to extend the industrial chain. The textile and apparel industry are mainly concentrated in the provinces of Inner Mongolia, Ningxia, Sichuan, Chongqing, Tibet, and Shaanxi. Inner Mongolia, Ningxia, and Tibet have advantages in terms of natural conditions for textiles and clothing, and Shaanxi and other places have a long history of textile production. The textile and apparel industry are also a light industry; investment is less than in heavy industries, but at the same time, it can generate a large number of jobs. Therefore, when various factors are considered, it is clear that there is support for these industries in these provinces.

**Table 6.** Summary of Foreign Trade Demonstration Bases in Western Region in 2018.

| Provinces | Number of Bases | Foreign Trade-Related Goods | Industries |
|---|---|---|---|
| Sichuan | 5 | Vegetables | Agricultural products |
| | | Women's shoes | Light industrial products |
| | | Apparel | Textile and Apparel |
| | | Non-ferrous metal materials | New Materials |
| | | Functional polymers and composites | Professional chemical |
| Ningxia | 2 | Cashmere products | Textile and Apparel |
| | | Wolfberry products | Pharmaceutical |
| Shaanxi | 3 | Apples | Agricultural products |
| | | Textile | Textile and Apparel |
| | | Non-ferrous metal materials | New Materials |
| Guangxi | 4 | Water products | Agricultural products |
| | | Ceramics | Light industrial products |
| | | Hangers | Light industrial products |
| | | Biomedicine | Pharmaceutical |
| Gansu | 2 | Apples | Agricultural products |
| | | Seeds | Agricultural products |
| Guizhou | 2 | Liquor | Agricultural products |
| | | New fertilizer | Specialized chemical industry |
| Inner Mongolia | 10 | Seed kernel | Agricultural products |
| | | Tomato | Agricultural products |
| | | Beef | Agricultural products |
| | | Multigrain beans | Agricultural products |
| | | Dehydrated vegetables | Agricultural products |
| | | Cashmere products | Textile and Apparel |
| | | Cashmere products | Textile and Apparel |
| | | Cashmere products | Textile and Apparel |
| | | Black metal material | New Materials |
| | | Black metal material | New Materials |
| Qinghai | 2 | Tibetan blanket | Agricultural products |
| | | Berries and products | Pharmaceutical |
| Xinjiang | 1 | Canned fruits and vegetables | Agricultural products |
| Yunnan | 3 | Flowers | Agricultural products |
| | | Vegetables | Agricultural products |
| | | Vegetables | Agricultural products |
| Chongqing | 5 | Mustard | Agricultural products |
| | | Lemon | Agricultural products |
| | | Beef | Agricultural products |
| | | Textile | Textile and Apparel |
| | | Western medicine | Pharmaceutical |

Source: Website of the Ministry of Commerce of the People's Republic of China.

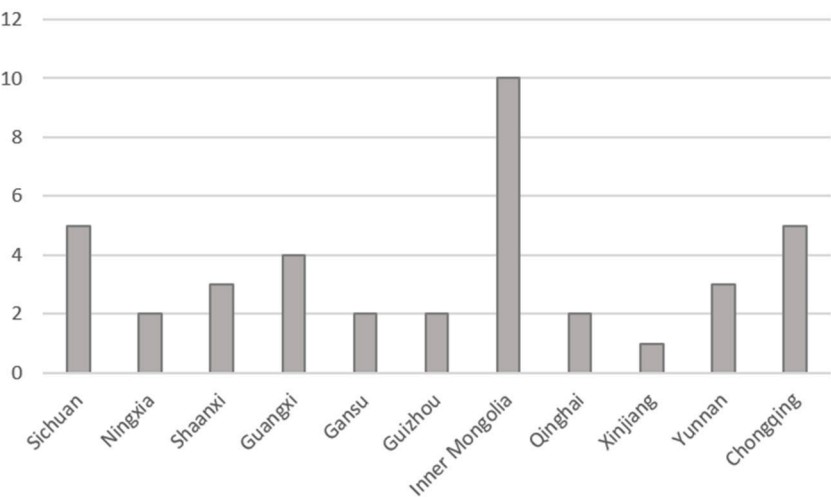

**Figure 1.** Number of Demonstration Bases for Foreign Trade in the Western Region. Note: The horizontal axis is the western provinces, and the vertical axis is the number of demonstration bases for foreign trade.

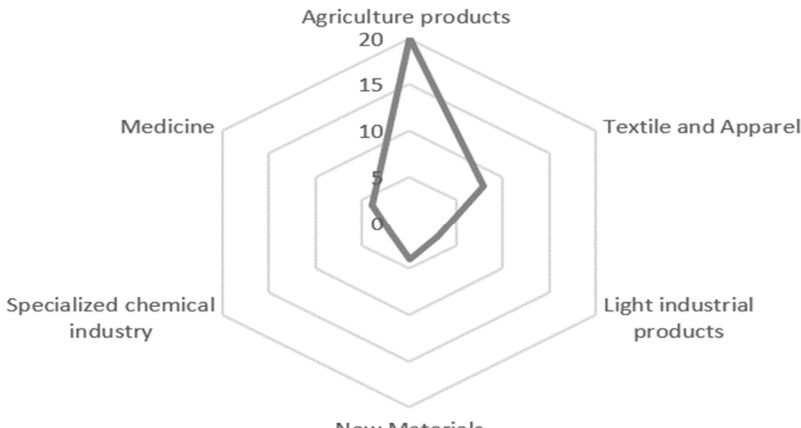

**Figure 2.** The Field of demonstration bases for foreign trade in the western region in 2018.

## 5. Conclusions

The main conclusions of this paper will now be given. The export of primary products and labor-intensive products contributes to the expansion of opening-up in the western region, and the latter plays a more significant role. Due to the slow development of capital-intensive industries, they have not reached the effect of a scale economy and agglomeration, so they have a restraining effect on the impact of the region's opening up to the outside world. Furthermore, the opening-up of the western region is significantly beneficial to the regional economic development, and on the basis of the opening up to the outside world, the export of labor-intensive products and capital-intensive products can significantly promote the regional economic development. Based on the above conclusions, this paper argues that western China should continue to expand foreign exchanges and expand the breadth and depth of opening up. Additionally, it should further optimize the foreign trade structure, focus on the development of primary products, and continue to promote the development of the foreign trade transformation demonstration base. According to the comparative advantages of local industrial development, special foreign trade industries should be developed, such as apple, textile, non-ferrous metal materials, and other industries in Shaanxi; cashmere and wolfberry products in Ningxia and so on. Combined with regional characteristics, the industrial chain should be extended to develop high value-added regional specialty products. In the development of labor-intensive products, we should continue to give full play to the advantages of the local labor force and gradually raise the skill level and quality of the local labor force on this basis to promote the smooth transformation and upgrading of local industries. In the development of capital-technology-intensive products, the western region should increase its attractiveness to capital and finance, strengthen the construction of high-tech parks, and pay more attention to keeping talents in the region on the basis of the training of talents in the region. The phenomenon of the outflow of talents from western regions to central and eastern regions has been continuing. Therefore, the government should attach importance to retaining talents and making them contribute to the economic development of the region. At the same time, the government should increase the attraction of foreign talents and provide a preferential and convenient environment for high-quality talents.

**Supplementary Materials:** The following are available online at http://www.mdpi.com/2227-7099/7/2/56/s1.

**Author Contributions:** Conceptualization, X.L.; Data curation, L.S. and M.J.; Formal analysis, X.L. and R.K.; Methodology, N.L.; Software, N.L., L.S., X.L. and M.J.; Writing—original draft, N.L., L.S., X.L., R.K. and M.J.

**Funding:** This research was funded by Projects of the National Social Science Foundation of China, and the subject is the dilemma of global climate change negotiations and research on China's active response to participation in negotiations, grant number 16BZZ086.

**Conflicts of Interest:** The authors declare no conflict of interest.

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
