# Peer review of "Foreign Trade Structure, Opening Degree and Economic Growth in Western China"

_economies, doi:10.3390/economies7020056_

Round 1

Reviewer 1 Report

The article has a low integrative value in the current mainstream research, it is badly edited, and it is quite poorly written in English. I question some of the statistical work, primarily for lack of clarity and less than thorough explanations. The quality of the manuscript makes the reported research unclear, both in its design as well as the result. There are passages that invoke previous work in problematic (though not critical) ways. The introduction section requires more discussion of the specific related research which would demonstrate the gap your research is filling. The results as they are currently presented appear superficial and more context and discussion of the results is required.

Author Response

Dear editor,

Thank you very much for taking the time to review our article; and further put forward many very constructive suggestions to help us improve our paper. We agree with you and modestly accept all your suggestions. Based on your suggestions, we have substantially revised the article. We will now present the modifications. Thank you for your support and help. The details of the modifications are as follows.

Point 1: I question some of the statistical work, primarily for lack of clarity and less than thorough explanations.

Response 1: Firstly, we have added new control variables and revised the empirical results in section 4.2 to ensure the empirical results’ scientificity. Furthermore, we have supplemented the explanations of part 4.2’s empirical results, improved the results from both statistical and economic perspectives, and supplemented the possible causes for this result.

Point 2:The quality of the manuscript makes the reported research unclear, both in its design as well as the result.

Response 2: In the research design, we have added part 4.3. At the beginning, this study analyzed the impact of trade structure on opening up(section 4.1) and the role of opening up on economic growth(section 4.2), respectively. Through the analysis of trade structure and opening-up, the final aim of this paper is to explore how the western region of China can promote economic development with a higher quality. This paper holds that the positive role of opening up to the outside world in promoting economic growth is closely related to the change of trade structure, and under the implementation of the policy of opening up to the outside world, the effect of the change of trade structure on economic growth is worth paying attention to. Therefore, this paper attempts to discuss the interaction between opening up and trade structure on economic growth. Therefore, on the basis of the dynamic impact of opening up on economic growth, this paper introduces the interaction item of foreign trade structure in order to clearly grasp the relationship between the three variables. This is why part 4.3 was added to here. Similarly, the explanations of the regression results in part 4.3 have been supplemented with statistical and economic meanings.

Because of the change to the empirical part, this paper has included a new descriptive analysis for part 4.4. In this part, based on the empirical results, this paper reorganizes and summarizes the relationship between foreign trade structure, opening up and economic growth, and discusses the possible reasons for the results.

The conclusion (part 5) of this paper has been reinterpreted. Based on the previous empirical results, each piece of information is summarized. According to each conclusion, relevant policy suggestions are put forward, thus making the conclusion part more logical and clear.

Point 3:The introduction section requires more discussion of the specific related research which would demonstrate the gap your research is filling.

Response 3: The introduction (part 1) of this paper has been extended, a background introduction to the western region development has been added, and a comparison of the eastern and western regions has been included, which leads to the reasons why this paper now focuses on the development of the western region, and at the same time, emphasizes the contributions made by this paper.

    In terms of the literature review, we have increased the collation about domestic and foreign literature, expanded the amount of literature referenced, and emphasized the significance of this paper as well as further clarified the purpose of this study, which provides better proof for the gaps filled by this research.

Point 4:The results as they are currently presented appear superficial and more context and discussion of the results is required.

Response 4: In the final conclusion (part 5), the conclusions have been re-summarized, and more discussion on the conclusions in light of the above has been added. Each conclusion is based on the previous empirical results, which makes the content of the conclusion more detailed and clear. Furthermore, the policy recommendations are based on each conclusion, which improves the logic of the conclusions

Point 5: It is badly edited, and it is quite poorly written in English.

Response 5: This paper entrusts the MDPI platform to edit and correct the grammar and language style. MDPI is a professional language editing platform. Through its help, the writing of the article in terms of the structure and content has been made more professional and standardized.

Point 6: Methods adequately described must be improved.

Response 6: In part 3.1, the research methods have been complemented and improved. On the basis of opening up, we have constructed a dynamic model including foreign trade structure, opening up and economic growth. Specifically, in terms of the dynamic impact model of opening up for economic growth, we have added the interaction item of foreign trade structure and opening up. Since the former adopts the system GMM model, and the interaction term is added on the basis of the former, the dynamic model is also constructed here for identification.

In addition, the table of empirical results presented in the paper was reedited, the results without control variables were combined, and the results after adding control variables were combined, so as to make the article look more standardized and readable. Due to the content changes, the abstract has also been improved. The revised article is 51% longer than the original version, and the structure and content have been improved, making the whole article more standardized and full.

Thank you very much for your patient review. 

Reviewer 2 Report

-Language should be improved.

-Authors benefit from panel techniques and find an interesting results. 

-Specially labor intensive goods exports are found to be beneficial for China. 

-Paper has a contribution in terms of getting into details of trade dynamics for China. 

Author Response

Dear editor,

Thank you very much for taking the time to review our article; and further put forward many very constructive suggestions to help us improve our paper. We agree with you and modestly accept all your suggestions. Based on your suggestions, we have revised the article. We will now present the modifications. Thank you for your support and help. The details of the modifications are as follows.

Point 1:Language should be improved

Response 1:This paper entrusts the MDPI platform to edit and correct the grammar and language style. MDPI is a professional language editing platform.Through its help,the writing of the article in terms of the structure and content has been made more professional and standardized.

Point 2:Is the research design appropriate?——Can be improved

Response 2: In the research design, we have added part 4.3. At the beginning, this study analyzed the impact of trade structure on opening up(section 4.1) and the role of opening up on economic growth(section 4.2), respectively. Through the analysis of trade structure and opening-up, the final aim of this paper is to explore how the western region of China can promote economic development with a higher quality. This paper holds that the positive role of opening up to the outside world in promoting economic growth is closely related to the change of trade structure, and under the implementation of the policy of opening up to the outside world, the effect of the change of trade structure on economic growth is worth paying attention to. Therefore, this paper attempts to discuss the interaction between opening up and trade structure on economic growth. Therefore, on the basis of the dynamic impact of opening up on economic growth, this paper introduces the interaction item of foreign trade structure in order to clearly grasp the relationship between the three variables. This is why part 4.3 was added to here. Similarly, the explanations of the regression results in part 4.3 have been supplemented with statistical and economic meanings.

    Because of the change to the empirical part, this paper has included a new descriptive analysis for part 4.4. In this part, based on the empirical results, this paper reorganizes and summarizes the relationship between foreign trade structure, opening up and economic growth, and discusses the possible reasons for the results. Similarly, for the newly added content, the conclusion (part 5) has also been revised.

Point 3: Are the methods adequately described?——Can be improved

Response 3: In part 3.1, the research methods have been complemented and improved. On the basis of opening up, we have constructed a dynamic model including foreign trade structure, opening up and economic growth. Specifically, in terms of the dynamic impact model of opening up for economic growth, we have added the interaction item of foreign trade structure and opening up. Since the former adopts the system GMM model, and the interaction term is added on the basis of the former, the dynamic model is also constructed here for identification.

Point 4: Are the results clearly presented?——Can be improved

Response 4: The conclusion (part 5) of this paper has been reinterpreted. Based on the previous empirical results, each piece of information is summarized. According to each conclusion, relevant policy suggestions are put forward, thus making the conclusion part more logical and clear.

Point 5: Are the conclusions supported by the results——Can be improved

Response 5: In the final conclusion (part 5), the conclusions have been re-summarized, and more discussion on the conclusions in light of the above has been added. Each conclusion is based on the previous empirical results, which makes the content of the conclusion more detailed and clear. Furthermore, the policy recommendations are based on each conclusion, which improves the logic of the conclusions.

In addition, the table of empirical results presented in the paper was reedited, the results without control variables were combined, and the results after adding control variables were combined, so as to make the article look more standardized and readable. Due to the content changes, the abstract has also been improved. The revised article is 51% longer than the original version, and the structure and content have been improved, making the whole article more standardized and full.

Thank you very much for your patient review. 

Reviewer 3 Report

The paper analyses the relationship between foreign trade structure, opening degree and economic growth of the provinces in western China. The topic is interesting however, limited to a specific geographical area. The paper is well-structured. For the econometric analysis the paper uses GMM. As far as I can ascertain the econometric method has been used correctly implemented. The hypotheses presented are stated clearly. The paper reviews the current literature and connects it with empirical research. From an editorial point of view, this paper is of a publishable standard as is.

Author Response

Dear editor,

Thank you very much for taking the time to review our article; and further put forward many very constructive suggestions to help us improve our paper. We agree with you and modestly accept all your suggestions. Based on your suggestions, we have revised the article. We will now present the modifications. Thank you for your support and help. The details of the modifications are as follows.

Point 1: English language and style are fine/minor spell check required 

Response 1: This paper entrusts the MDPI platform to edit and correct the grammar and language style. MDPI is a professional language editing platform. Through its help, the writing of the article in terms of the structure and content has been made more professional and standardized.

Point 2: The topic is interesting however, limited to a specific geographical area.

Response 2: The introduction (part 1) of this paper has been extended, a background introduction to the western region development has been added, and a comparison of the eastern and western regions has been included, which leads to the reasons why this paper now focuses on the development of the western region, and at the same time, emphasizes the contributions made by this paper.

In addition, the table of empirical results presented in the paper was reedited, the results without control variables were combined, and the results after adding control variables were combined, so as to make the article look more standardized and readable. Due to the content changes, the abstract has also been improved. The revised article is 51% longer than the original version, and the structure and content have been improved, making the whole article more standardized and full.

Thank you very much for your patient review. 

Reviewer 4 Report

The paper covers an interesting topic. The analysis is well conducted and the results are interesting.

The literature review provides a good background for the empirical exercise.

My only small aspect concerns the introduction and motivation of the paper. In my opinion, it could be stronger.

Author Response

Dear editor,

Thank you very much for taking the time to review our article; and further put forward many very constructive suggestions to help us improve our paper. We agree with you and modestly accept all your suggestions. Based on your suggestions, we have revised the article. We will now present the modifications. Thank you for your support and help. The details of the modifications are as follows.

Point 1: English language and style are fine/minor spell check required

Response 1: This paper entrusts the MDPI platform to edit and correct the grammar and language style. MDPI is a professional language editing platform. Through its help, the writing of the article in terms of the structure and content has been made more professional and standardized.

Point 2: My only small aspect concerns the introduction and motivation of the paper. In my opinion, it could be stronger. 

Response 2: The introduction (part 1) of this paper has been extended, a background introduction to the western region development has been added, and a comparison of the eastern and western regions has been included, which leads to the reasons why this paper now focuses on the development of the western region, and at the same time, emphasizes the contributions made by this paper.

    In view of the clarity of the purpose of this study, in addition to expanding and supplementing the introduction, in terms of the literature review, we have increased the collation about domestic and foreign literature, expanded the amount of literature referenced, and emphasized the significance of this paper as well as further clarified the purpose of this study, which provides better proof for the gaps filled by this research.

     In the research design, we have added part 4.3. It has played a further supporting and improving role in the implementation of the purpose of this study. At the beginning, this study analyzed the impact of trade structure on opening up(section 4.1) and the role of opening up on economic growth(section 4.2), respectively. Through the analysis of trade structure and opening-up, the final aim of this paper is to explore how the western region of China can promote economic development with a higher quality. This paper holds that the positive role of opening up to the outside world in promoting economic growth is closely related to the change of trade structure, and under the implementation of the policy of opening up to the outside world, the effect of the change of trade structure on economic growth is worth paying attention to. Therefore, this paper attempts to discuss the interaction between opening up and trade structure on economic growth. Therefore, on the basis of the dynamic impact of opening up on economic growth, this paper introduces the interaction item of foreign trade structure in order to clearly grasp the relationship between the three variables. This is why part 4.3 was added to here. Similarly, the explanations of the regression results in part 4.3 have been supplemented with statistical and economic meanings. In this part, this paper re-emphasizes the foothold of the article, in order to clarify the motivation of this study.

    Because of the change to the empirical part, this paper has included a new descriptive analysis for part 4.4. In this part, based on the empirical results, this paper reorganizes and summarizes the relationship between foreign trade structure, opening up and economic growth, and discusses the possible reasons for the results.

In addition, the table of empirical results presented in the paper was reedited, the results without control variables were combined, and the results after adding control variables were combined, so as to make the article look more standardized and readable. Due to the content changes, the abstract has also been improved. The revised article is 51% longer than the original version, and the structure and content have been improved, making the whole article more standardized and full.

Thank you very much for your patient review. 

Round 2

Reviewer 1 Report

The manuscript looks better now. There are some incomplete info in the reference list. Also, although you cover recent developments, the literature you use is mostly too old.

Author Response

Dear editor,

Thank you very much for taking the time to review our articleï¼›We agree with you and modestly accept all your suggestions. Based on your suggestions, we have revised the article. We will now present the modifications. Thank you for your support and help. The details of the modifications are as follows.

Point 1: There are some incomplete info in the reference list

Response 1: We have completed the information in the reference list.

Point 2:  The literature you use is mostly too old.

Response 2: We have added a lot of references in recent years and deleted the old ones. And the modification of the language has been completed.

Thank you very much for your patient review.
